The spatial and temporal evolution of habitat quality and driving factors in nature reserves: a case study of 33 forest ecosystem reserves in Guizhou Province

Mei Xuemeng
Liu Yi
Yue Li
Zhang Mingming mmzhang@gzu.edu.cn
1 Guizhou University, College of Forestry , Guiyang , Guizhou , China
2 Research Center for Biodiversity and Nature Conservation of Guizhou University , Guiyang , China
Root-Bernstein Meredith
Electronic publication date: 2025 Mar 24
Publication date: 2025
Volume: 13
Electronic Location ID: e19098
Received 2024 Sep 20; Accepted 2025 Feb 12
Copyright: ©2025 Mei et al.
Copyright year: 2025
Copyright holder: Mei et al.
License: This is an open access article distributed under the terms of the Creative Commons Attribution License, which permits unrestricted use, distribution, reproduction and adaptation in any medium and for any purpose provided that it is properly attributed. For attribution, the original author(s), title, publication source (PeerJ) and either DOI or URL of the article must be cited.
License URL: https://creativecommons.org/licenses/by/4.0/

Keywords: Land use change, Habitat quality, InVEST model, Optimal parameters-based geographical detector, Driving factors, Nature reserves

Funding: National Natural Science Foundation of China 32260331 This research was funded by the National Natural Science Foundation of China (Grant Number 32260331). The funders had no role in study design, data collection and analysis, decision to publish, or preparation of the manuscript.

==============================
Background

Biodiversity plays a crucial role for humanity, serving as a foundation for human survival and development. Habitat quality serves as a critical indicator for assessing biodiversity and holds significant importance in both theoretical and practical domains. The unique natural geographical environment of Guizhou Province has fostered rich biodiversity and facilitated the establishment of numerous nature reserves, predominantly centered on forest ecosystems. Analyzing the habitat quality of nature reserves and its influencing factors is of great significance for maintaining the regional ecosystem stability, promoting sustainable development, and improving the ecological environment.

Method

Therefore, taking the 33 nature reserves of forest ecosystem in Guizhou Province as the study area, we first quantified habitat quality using the Integrated Valuation of Ecosystem Services and Trade-offs (InVEST) model to analyze changes in the nature reserve from 2000 to 2020. Then, we explored the effects of natural and social factors on the spatiotemporal evolution of habitat quality using the optimal parameters-based geographical detector (OPGD).

Results

Forests were identified as the primary land-use type in the study area. However, the nature reserves saw an increase area in cropland, and impervious land by 5,001.39 ha and 102.15 ha; a significant decrease in forests and grasslands; and a slight decrease in watersheds. Rapid urbanization, therefore, negatively affected the overall habitat quality of the reserve. Although there is a declining trend in the habitat quality of the nature reserve, the magnitude of change from 2010 to 2020 (−0.04) is smaller than that from 2000 to 2010 (−0.17), indicating that the management of the reserve has been somewhat effective. In national-level nature reserves, interactions between natural geographic factors and socio-economic factors were greater than interactions between natural geographic factors. Similarly, in local-level nature reserves, interactions between natural geographic factors and socio-economic factors outweighed interactions among social factors.

Conclusion

The spatiotemporal variability of habitat quality in the study area was shaped by the combined effects of natural and social factors. The habitat quality of local-level protected areas is, furthermore, more significantly affected by human activities, which are the primary cause of their degradation.

Introduction

Habitat quality refers to the capacity of the natural environment to support the survival of individual species and populations by providing all essential conditions (Krausman, 1999; Song et al., 2020a). It also serves as an indicator of the ecosystem services and biodiversity of a specific area (Sallustio et al., 2017). However, rapid urbanization and associated land-use changes have resulted in biodiversity loss, significant reduction in ecosystem services, and severe degradation of biological habitats. Consequently, biodiversity conservation faces substantial challenges (Lin et al., 2024; Wu, Sun & Fan, 2021). Protected areas have been shown to be effective for in situ biodiversity conservation (Yan et al., 2024). Recent assessments reveal that approximately one-third of the world’s nature reserves are experiencing high levels of pressure, and 55% have seen an increase in anthropogenic influence (Jones et al., 2018). Land-use change is a critical indicator of human activity and a significant risk factor for the quality of natural environments and biological habitats (Lin et al., 2024; Zhang, Jiangling & Mengjie, 2024). In particular, urban development and agricultural expansion have intensified the destruction of these environments, resulting in habitat loss (Fang et al., 2021; Zhao et al., 2022). The investigation of spatial and temporal changes in habitat quality and the analyses of factors influencing these changes, therefore, are crucial for sustainable development and for enhancing the protection of biodiversity in nature reserves.

Habitat quality evaluation methods primarily consist of an indicator system evaluation and an ecological model assessment (Watson et al., 2016). An increasing number of researchers are adopting model-based assessments due to the extensive data requirements associated with the indicator system evaluation (Zhao et al., 2022). The Integrated Valuation of Ecosystem Services and Trade-offs (InVEST model) stands out for its ease of data collection, high accuracy, and robust spatial representation, and is widely adopted in various studies (Zhao, Li & Liu, 2022). For example, the InVEST-HQ model has been used to assess dynamic changes in habitat quality in China (Chen & Liu, 2024; Li et al., 2024; Zhang et al., 2022; Zhao et al., 2022), Portugal (Moreira et al., 2018) and Scottish (Fabris, Buddendorf & Soulsby, 2019).

The methodology and the indicator system of the InVEST-HQ model have been continually refined through multiple validations. Overall, both local and international researchers have conducted extensive and multi-scale studies on habitat quality assessment. However, there is a lack of research on the spatial and temporal changes of regional-level protected forest ecosystem quality, particularly in terms of physical-geographical and socio-economic dimensions.

The optimal parameter-based geographical detector (OPGD) model identifies the interaction between two factors and the dependent variable by calculating and comparing the q-values of individual factors and their combined effects. It also determines whether the intensity and direction of these interactions are linear or nonlinear, thereby uncovering complex local relationships from a spatial perspective, effectively addressing the limitations of traditional research methods (Song et al., 2020b; Wang et al., 2024). Researchers have begun utilizing the OPGD model to investigate the effects of human activities, climate change, and other factors on ecosystem services and land use dynamics. Gu et al. (2024) utilized the OPGD model to quantify the effects of topography, climate, and human activities on the spatial distribution of vegetation growth, as indicated by kNDVI. Similarly, Cen, Zhou & Qiu (2024) integrated MGWR and OPGD to assess the influence of natural and anthropogenic factors on the spatial quality patterns of waterfront urban neighborhoods. However, few studies have employed the OPGD model to analyze the spatiotemporal evolution of habitat quality in nature reserves under the influence of natural and social factors.

Guizhou Province, situated in the eastern part of China’s southwestern region, lies in the heart of the Yunnan-Guizhou Plateau. It is strategically located in the upper reaches of both the Yangtze River and the Pearl River, serving as a critical ecological barrier for these watersheds. Additionally, Guizhou is recognized as a National Ecological Civilization Pilot Zone (Niu & Shao, 2020; Wang, Zhang & Chen, 2022). Over the years, leveraging its rich resource base, Guizhou has progressively enhanced its biodiversity protection framework, establishing numerous nature reserves in areas with concentrated distributions of protected species and landscape resources. National and local nature reserves serve key roles in conservation but differ significantly. National reserves, managed by the central government, provide higher protection levels, focusing on ecosystems or endangered species of ecological significance. These reserves benefit from ample funding, research resources, and strict management regulations. In contrast, local reserves, managed by provincial, municipal, or county governments, prioritize region-specific ecosystems or species. Typically smaller in scale, they face resource constraints and limited influence, resulting in more flexible protection measures tailored to local priorities. Current research on habitat quality in Guizhou Province primarily examines the impact of individual factors, such as topography, land use, and urban expansion, within relatively continuous and intact regions. For example, Xie & Zhang (2023) analyzed the spatial and temporal changes in habitat quality in Guizhou Province from 1990 to 2018 and used the geodetector model to identify the factors influencing habitat quality. Zijin, Xuesong & Mingman (2023) conducted a long-term study on the evolution of ecosystem services in mountainous karst areas of the Guizhou Province, focusing on both horizontal and vertical spatial dimensions. Huang et al. (2018) not only examined the effects of land-use change on habitat quality across different functional zones but also quantitatively assessed how converting farmland to forest and grassland improved habitat quality in the Pogang Nature Reserve, Xingyi City, Guizhou Province. However, there is still a notable lack of research on the spatial and temporal characteristics of habitat quality and the multifactorial mechanisms of influence on the nature reserves in Guizhou Province. Furthermore, most current research is mainly performed at the small to medium scales, such as individual rivers (Duffin et al., 2023; Xu et al., 2019) and specific nature reserves (Xu et al., 2023), with limited attention to broader changes across multiple reserves. Additionally, few studies include a comprehensive analyses of factors such as natural conditions, landscape patterns, social development, and economic growth when analyzing habitat quality.

To address the identified research gap, this study focuses on 33 forest ecosystem nature reserves in Guizhou Province. Initially, habitat quality and its changes from 2000 to 2020 were quantified using the InVEST model. Subsequently, potential influencing factors were selected from both natural geographic and socioeconomic dimensions, and the OPGD model was employed to assess the impact of these factors on habitat quality. The main objectives of this study are to: (1) reveal the spatiotemporal dynamics of land use and habitat quality within nature reserves; and (2) identify key factors affecting habitat quality changes, analyze their interactions, and elucidate underlying mechanisms impacting habitat quality in these reserves, thereby providing a reference for scientifically informed reserve management.

Materials and Methods

Study area

According to the list of nature reserves in Guizhou Province published by the Guizhou Forestry Bureau, as of 2018, there are 11 national-level and 89 local-level nature reserves in Guizhou. The majority of these reserves are forest ecosystem types. Considering the integrity and stability of ecosystems, we excluded protected areas that were too small to sustain complete forest ecosystems. We prioritized larger protected areas that exhibited higher biodiversity, contained habitats for endangered species, or possessed significant ecological functions, this study focuses on 33 nature reserves of forest ecosystem in Guizhou Province (Table 1 and Fig. 1) to analyze changes in habitat quality from 2000 to 2020. Each reserve will be assigned a code to facilitate the analysis.

Table 1 List of forest ecosystem nature reserves in Guizhou Province.

Number	Name of nature reserve in Guizhou	Code	Area (ha)	Establishment year	
1	Leigongshan National Nature Reserve, Guizhou Province	N1	47,300	1982	
2	Xishui National Nature Reserve in Guizhou Province	N2	51,911	1994	
3	Guizhou Fanjingshan National Nature Reserve	N3	41,907.99	1978	
4	Guizhou Chishui alsophila national Nature reserve	N4	13,900	1984	
5	Guizhou Dashahe National Nature Reserve	N5	26,990	1984	
6	Guizhou Kuankuoshui National Nature Reserve	N6	26,231	1989	
7	Guizhou Maolan National Nature Reserve	N7	24,891.4	1986	
8	Guizhou Foding Mountain National Nature Reserve	N8	15,200	1992	
9	Congjiang County moon Mountain state level nature reserve	R1	24,800	1992	
10	Rongjiang County moon Mountain state nature reserve	R2	32,849	1998	
11	Yinjiang County Yangxi provincial nature reserve	R3	21,871.48	2000	
12	Taijiang County Nangong state nature reserve	R4	22,104	2001	
13	Guizhou Meitan Baidianshui Provincial Nature reserve	R5	19,173	2001	
14	Baiqing, Huanglian nature reserve, Tongzi county	R6	23,007.96	1985	
15	Liping County Taiping Mountain state level nature reserve	R7	31,551	1989	
16	Sinan county Siyetun provincial nature reserve	R8	17,400	1999	
17	Jianhe county Baili broadleaf forest state nature reserve	R9	12,785.3	1997	
18	Huangping County Shangtang Zhujiashan state nature reserve	R10	12,346.17	1984	
19	Majiang County old snake chong state nature reserve	R11	8,678	1992	
20	Suiyang County Houshui River nature reserve	R12	7,500.22	2000	
21	Yinjiang county Shijialing nature reserve	R13	8,915.75	2000	
22	Xianren Mountain Nature Reserve	R14	6,733.33	2004	
23	Changshun Douma karst forest nature reserve	R15	3,333.3	2000	
24	Huishui Chang ’an Karst Nature Reserve	R16	4,313.09	2000	
26	Xingren Qingshui River scenic forest state level protection area	R17	2,556	1997	
26	Mafulin nature reserve of Wuchuan County	R18	3,710.3	2001	
27	Suiyang County cedar cyanine nature reserve	R19	2,400.29	2000	
28	Suiyang County Huoqiuba white crown long tailed pheasant nature reserve	R20	2,466.24	2000	
29	Zunyi County Sunjialin nature reserve	R21	3,003.1	2004	
30	Shibing County Foding mountain county nature reserve	R22	2,133.37	1994	
31	Jinping County Eight Rivers County Nature Reserve	R23	4,200	1998	
32	Cloak Mountain nature reserve	R24	2,672	1998	
33	Zhenfeng Longtongshan water source forest state level protection area	R25	2,817	1997	
Notes.

‘N’ represents national-level nature reserves, while ‘R’ signifies local-level nature reserves.

Figure 1 Regional location of 33 forest ecosystem types nature reserves in Guizhou Province.

The selected 33 nature reserves in this study are primarily located in the northern and southeastern regions of Guizhou (Fig. 1). These areas face the simultaneous challenges of economic development and ecological protection, leading to a particularly pronounced conflict between environmental conservation and poverty alleviation efforts. Additionally, these regions have recently become major tourism hotspots. For example, the Xijiang Miao Village and the Fanjing Mountain scenic area are situated adjacent to the Leigong Mountain National Nature Reserve and the Fanjing Mountain National Nature Reserve, respectively, intensifying the conflict between economic development and ecological protection.

Data resources

The data used in this study primarily includes GIS spatial data as inputs for the model. The threat factors consist of two types: cropland and construction land. Moreover, the land use data for the years 2000, 2010, and 2020 were obtained from the China Land Cover Annual Dataset (CLCD) (https://doi.org/10.5281/zenodo.5816591), which provides raster data with a spatial resolution of 30 m. Furthermore, in this study, the land use types in the study area were categorized into seven types: forest, farmland, shrub, grassland, water bodies, impervious surfaces, and wasteland.

Additional data, including the digital elevation model (DEM), average annual temperature, average annual precipitation, population density, and gross domestic product (GDP), were obtained from the Data Center for Resource and Environmental Sciences of the Chinese Academy of Sciences (http://www.resdc.cn). Slope and elevation values were extracted from the DEM using ArcGIS 10.8. All data were sourced from official platforms as detailed in Table 2. Moreover, Fig. 2 presents a schematic representation of our research methodology.

Table 2 Research data and sources.

Category	Name	Format	Time	Source	
Land use data	Land use data	Raster data (30 m)	2000, 2010, 2020	Annual China Land Cover Dataset, CLCD (https://doi.org/10.5281/zenodo.5816591)	
Natural environment data	Annual Average Temperature	Raster data (1 km)	2000, 2010, 2020	Data Center for Resources and Environmental Sciences, Chinese Academy of Sciences (http://www.resdc.cn)	
Annual Average Precipitation	Raster data (1 km)	2000, 2010, 2020	
Basic geographic data	Digital Elevation Model	Raster data (90 m)	–	
Research Area Boundary	Image	2018	Guizhou Provincial Forestry Bureau	
(https://lyj.guizhou.gov.cn/, accessed on 20 December 2023)	
Socio-cultural data	GDP	Raster data (1 km)	2000, 2010, 2020	Data Center for Resources and Environmental Sciences, Chinese Academy of Sciences (http://www.resdc.cn)	
Population density	Raster data (1 km)	2000, 2010, 2020	

Figure 2 Flowchart of the research methodology.

The landscape pattern index

Markov models serve as the foundation for the Land use transfer matrix (Jawarneh et al., 2024). In a recent study, changes in land-use types between the start and end of the study period were determined, and the original sources of each land-use type at the end of the study period were identified using the Markov model (Feudjio Fogang et al., 2023). The Markov model is, therefore, suitable for the assessment of land-use changes (Ait El Haj, Ouadif & Akhssas, 2023; Congalton, 1991; Dietzel & Clarke, 2006). The formula for land-use change is as foll (1) Sij=S11S12⋯S1nS21S22⋯S2nSn1Sn2⋯Snn

where Sij represents the area of land initially classified as type i and converted to type j by the end of the study, and n denotes the total number of land-cover types.

The dynamics of integrated land use reflect the overall speed and magnitude of land-use changes within the study area. A higher value of integrated land-use dynamics indicates a faster rate of integrated land-use change in the region. (2) Lc=∑i=1nΔLUi−j2∑i=1nLUI×1T×100%

where LC represents the integrated land-use dynamics, LUi denotes the area of category i land-use type at the beginning of the study, and ΔLUi−j denotes the absolute value of the area of the i land-use type that was converted to a non-i land-use type during the study period.

The InVEST HQ model and spatial autocorrelation analysis

The InVEST HQ model

Habitat quality refers to the capacity of an ecosystem to support the sustainable development of both individual organisms and populations (Janus & Bozek, 2019; Yang, 2021). The InVEST Habitat Quality model, developed by Stanford University, utilizes land use and land cover change (LUCC) data from the study area to assess the impacts of threat factors on various land use types and their sensitivities, thereby characterizing regional habitat quality (Wu et al., 2019). Typically, the severity of a threat diminishes with increasing distance between the grid cells and the threat source. Consequently, those grid cells closest to the threat are often subjected to greater impacts. The intensity of different threat factors on habitat grid cells decreases as the distance from the threat source increases, which can be quantitatively described using the following equation (Liu et al., 2019; Sharp et al., 2018): (3) irxy=1−dxydrmaxlineardecayexp−2.99drmax×dxyexponentialdecay

where irxy represents the influence of the grid y covered by the threat factor r on the grid covered by habitat type j; d denotes the distance between grid x and grid y; and dxy indicates the maximum distance of the threat factor. If irxy > 0, it indicates that grid x is located within the disturbance area of grid y.

For a comparative analysis, in the present study, the habitat quality index was categorized into four intervals using an equal-interval grading method: low (0–0.25), medium (0.25–0.50), moderately high (0.50–0.75), and high (0.75–1.00) (Li et al., 2024). The formula used is as follows (Yu-Bin et al., 2015): (4) Qxj=Hj×1−Dxjz/Dxjz+kz

where Qxj represents the habitat quality of raster x for land use and land cover (habitat type) j, Hj denotes the habitat suitability of land use and land cover j, z is a default parameter of the model, and k is the half-saturation constant, set as half of the maximum value of Dxj.

In this study, the setup process was informed by both the user guide of the InVEST model and previous studies (Goldstein et al., 2012; Nelson et al., 2010; Terrado et al., 2016). The specific setup parameters are detailed in Tables 3 and 4.

Table 3 Habitat quality threat source attributes.

Threats factors	Max distance of influence/km	Weights	Decay type	
Cropland	6	0.7	Linear	
Construction land	5	1	Exponential	

Table 4 Sensitivity of different habitat types to threat factors.

Land use type	Habitat suitability	Cropland	Construction land	
Cropland	0.5	0	0.6	
Forestland	1	0.7	0.65	
Shrubland	1	0.65	0.6	
Grassland	0.7	0.55	0.5	
Watershed	0.8	0.6	0.4	
Uncultivated land	0.2	0	0	
Construction land	0	0	0	

Spatial autocorrelation analysis

Spatial autocorrelation was analyzed using the global Moran’s I index to assess whether the distribution of habitat quality in protected areas exhibited a clustered pattern (Jinrui et al., 2019). Analyses of hotspots and coldspots aim to identify and quantify the degree of clustering and the statistical significance in geospatial data. Hotspots and coldspots represent clusters of high and low values, respectively, which are visualized by this method. Spatial distribution patterns and aggregation characteristics of the data can thereby be revealed (Song et al., 2020b; Xiao-Han & Xiu-Juan, 2024). The formulae used are as follows: (5) I=n∑i=1n ∑j=1nwijxiiX ¯xjX ¯∑i=1nxiX ¯2∑i=1n ∑j=1nwij

(6) Gi∗=∑j=1nwijxj− ∑j=1nwijS∑j=1nwijxj−∑j=1nwij2n=1

(7) X ¯=∑j=1nxjn

(8) S= ∑j=1nxj2n−X¯2

where I refers to Moran’s index; Gi∗ is the hotspot index; wij represents the spatial weight between the i and j spatial cells; xi and xjdenote the values of the i and j cells, respectively; X ¯ represents the mean value of the cells; and n is the total number of cells in the study area. Moran’s I index ranges between −1 and 1, with larger values indicating a stronger spatial correlation, values less than 0 indicating a negative correlation, and a value of 0 indicating a random distribution.

Optimal parameters-based geographical detector

In this study, seven indicators were selected from both natural environmental and socioeconomic dimensions to investigate the main factors influencing the habitat quality of forest ecosystem nature reserves in Guizhou Province, and the evolutionary mechanisms, taking into account the specific characteristics of these reserves (Table 5). Topography, a fundamental natural environmental factor, forms the basis for the development and evolution of geographic environments (Wu, Sun & Fan, 2021). In this study, elevation and slope were used to represent topography. Precipitation and temperature, which influence plant growth and development, were represented by average annual precipitation and annual temperature. Land-use intensity reflects effects of human activity and was categorized based on the degree of environmental disturbance: forests, shrubs, grasslands, and water bodies were classified as Category 1 and represent a relatively natural state; agricultural land was classified as Category 2; and impervious surfaces were classified as Category 3. Agricultural land and impervious surfaces typically suggest more intensive human activities, often manifesting as land modification, vegetation destruction, and habitat degradation, all of which can severely affect ecosystems. In contrast, Category 1 areas, such as forests and water bodies, may offer better ecosystem protection. Gross Domestic Product (GDP) serves as a key indicator of the economic status and developmental level of a region. Population density provides information on the distribution of people within a unit area. By selecting GDP and population density, both the degree of human activity and economic transactions (e.g., tourism) affecting nature reserves can be captured. Higher levels of disturbance exert greater pressure on the protected areas, leading to ecosystem degradation, which in turn negatively affects species and habitat quality within these reserves.

Table 5 Table of drivers of habitat quality change in nature reserves.

	Category	Element	Code	Factor	Calculation method and dimension	
Dependent variable	Habitat quality	Habitat quality	Y1	Habitat quality in 2000	–	
Y2	Habitat quality in 2010	–	
Y3	Habitat quality in 2020	–	
Independent variable	Natural geography	Topography	X1	Elevation	DEM data extraction (m)	
X2	Slope	Elevation difference/water	
Climatic Conditions	X3	Annual Average Temperature	°C	
X4	Annual Average Precipitation	mm	
Socio-cultural	Indigenous activities	X5	Land use intensity	Forest,shurb,grassland,water = 1	
Cropland = 2 Impervious = 3	
X6	GDP	indicators of economic conditions Billion/yuan	
X7	Population Density	Person per hectare	

Traditional geographic detectors require manually setting up the parameters when discretizing continuous variables, leading to issues of subjectivity and suboptimal discretization. To overcome these limitations, the Optimal Parameters-based Geographic Detector (OPGD) model was used to analyze the drivers of spatial and temporal changes in habitat quality of the forest ecosystem nature reserves in Guizhou Province (Dan-Dan & Yong-liang, 2023; Song et al., 2020b).

First, the explanatory power (q value) of each continuous-type driver is calculated under different classification methods and varying numbers of classifications. The q value ranges from 0 to 1, with a higher q value indicating a stronger explanatory power and a lower q value indicating a weaker explanatory power of the driver for spatial distribution of disaster sites. The OPGD model selects the optimal discretization method and number of breaks for each variable based on the maximum q value (Cen, Zhou & Qiu, 2024), thereby ensuring the highest possible explanatory power. The two-way interaction detector evaluates whether the combined effect of two factors enhances or weakens the explanatory power of the dependent variable Y (Yang et al., 2024). This analysis was conducted using the “GD” package in RStudio.

The interaction detector evaluates whether the combined effect of two influencing factors enhances or reduces the explanatory power of the dependent variable Y. The relationship between the two factors is presented in Table 6.

Table 6 Types of dual-factor interactions.

Judgment type	Type	
q(X1∩X2) < min(q(X1), q(X2))	Nonlinear diminishing	
min(q(X1), q(X2)) < q(X∩X2) < max(q(X1), q(X2))	Single-factor nonlinear diminishing	
q(X1∩X2) > max(q(X1), q(X2))	Dual-factor enhancement	
q(X1∩X2) = q(X1) + q(X2)	Independent	
q (X1∩X2) > q(X1) + q(X2)	Nonlinear enhancement	

Results

Land-use change analysis

To investigate the land use type transitions within the nature reserves of forest ecosystem in Guizhou Province over the past two decades, this study conducted a comparative analysis of land use data from two periods: 2000 to 2010 and 2010 to 2020. Corresponding land use transition matrices were constructed (Table S1). The analysis of these continuous time period matrices revealed the patterns of land use evolution in the Guizhou reserves.

The land use transition and chord diagrams (Fig. 3) indicate that forests dominate the study area, covering approximately 90% of the total area. A ranking of land use transitions by area across the two time periods reveals that over the past 20 years, land use transitions in the study area predominantly involved conversions from forest to agricultural land, forest to impervious surfaces, and forest to shrubland. Specifically, from 2000 to 2010, a total of 5,518.89 hectares of land experienced transition, characterized by the predominant conversion of forest to agricultural land and shrubland, with minimal transition to grassland. The increase in impervious surfaces primarily resulted from the conversion of agricultural land. In the subsequent period from 2010 to 2020, shrubland transitioned mainly back to forest, with a smaller portion converting to agricultural land. Forests continued to primarily transition to agricultural land and shrubland, but the outflow was less than the inflow. Most of the impervious surface increase originated from agricultural land, with a smaller contribution from forests, and the rate of increase was higher than during the 2000 to 2010 period.

Figure 3 (A–D) Land use transfer diagram of nature reserves in Guizhou Province.

The thickness of the line represents the amount of transfer.

The overall land use dynamics index rose from 0.06 in 2000 to 0.1 in 2020, indicating an acceleration in the rate of land use change and greater overall land use activity (Table 7). However, the rates of change varied among different land use types, with agricultural land, impervious surfaces, and water bodies exhibiting increased dynamics, while forests, grasslands, and shrublands showed decreased dynamics. The dynamic level of forests rose during the 2000 to 2010 period but declined in the 2010 to 2020 period, with a slowdown in the rate of area reduction. Meanwhile, the dynamic level of impervious surfaces increased from 2000 to 2010 but declined from 2010 to 2020, indicating a reduced rate of area increase. This suggests that during this period, efforts to protect and restore forests were strengthened, which aligns with the aforementioned land use analysis results.

Table 7 Land use dynamics of nature reserves in Guizhou Province, 2000–2020(%).

Time period	Cropland	Forest	Grassland	Impervious	Shrub	Water	Integrated land-use dynamics	
2000–2010	0.74	−0.06	−3.36	4.61	1.49	−0.31	0.06	
2010–2020	0.88	−0.03	−3.33	16.52	−1.78	13.32	0.06	
2000–2020	1.45	−0.09	−12.59	7.42	−0.58	5.58	0.10	

Distributional characteristics of habitat quality in nature reserves

Spatiotemporal variations in habitat quality within nature reserves

Based on land use and threat factor data, this study employed the habitat quality module of the InVEST model to assess habitat quality across 33 nature reserves of forest ecosystem from 2000 to 2020. Spatial data on habitat quality were obtained for the years 2000, 2010, and 2020, with habitat quality values of 0.7089, 0.5400, and 0.4911, respectively, with standard deviations of 0.1539, 0.1635, and 0.1657 (Fig. 4). These results indicate a downward trend in habitat quality, with the rate of decline gradually decreasing over time. This suggests that the spatial heterogeneity of habitat quality within the nature reserves has progressively increased, consistent with the results of the prior land-use dynamic analysis.

Figure 4 Changes in habitat quality of 33 nature reserves of forest ecosystem in Guizhou Province, 2000–2020.

The results of the equal interval classification method in ArcGIS 10.8 showed that, habitat quality was categorized into four grades, and a table presenting the proportion of area in each grade was generated (Table S2). Integrating the results from Table S2, it was found that in 2000, habitat quality within the study area was primarily categorized as Highest Level and Higher Level. By 2010, the area classified as Highest Level had decreased substantially, while Higher Level remained nearly unchanged, and Lower Level showed a marked increase. In 2020, the area with low-quality habitat continued to expand, though the rate of decline in Highest Level slowed significantly compared to the previous decade. Additionally, throughout the study period, habitat quality in national-level nature reserves generally remained high, with Higher Level habitat quality consistently dominant. In contrast, habitat quality in local-level nature reserves was relatively average, with a more pronounced rate of decrease in Highest Level habitat area. Overall, the habitat quality in the study area was moderate. However, the distribution of high-, medium-, and low-habitat-quality areas within different levels of the nature reserves was uneven. For example, in 2010, high-quality habitats accounted for only 10.36% of the total area, with the majority (10.13%) concentrated in national-level nature reserves. The only local-level reserves with high-quality habitats were Congjiang County Moon Mountain Nature Reserve (R1), Rongjiang County Moon Mountain Nature Reserve (R2), and Baiqing-Huanglian Nature Reserve in Tongzi County (R6), accounting for 0.23% collectively. The proportion of good-quality habitat was 55.70%, with national and local reserves contributing 28.01% and 27.69%, respectively. Poor and lowest-quality areas accounted for a combined 33.95%, with national and local reserves comprising 10.32% and 23.63%, respectively. From 2000 to 2020, the high-quality-habitat area showed a declining trend, while relatively-high-quality habitats generally increased. Simultaneously, the low- and relatively-low-quality habitats increased annually. This pattern of decreasing high-quality-habitat areas and increasing relatively-high-, relatively-low-, and low-quality-habitat areas aligns with the previously observed declining trend in the habitat quality index. From 2000 to 2020, the habitat quality of the 33 forest ecosystem nature reserves exhibited an overall declining trend, with average decreases of 0.1689 and 0.0489, respectively, and a deceleration in the rate of decline.

Analysis of hotspot and coldspot ratios

A spatial autocorrelation analysis (Moran’s I) was conducted using ArcGIS 10.8, to examine whether changes in habitat quality across the 33 studied nature reserves in Guizhou Province exhibited spatial aggregation from 2000 to 2020. Sixteen nature reserves passed the significance test in 2000, 2010, and 2020, with global Moran’s I Z-scores exceeding 1.65 and p-values less than 0.1. Among these, eight reserves had Z-scores greater than 2.58 and p-values less than 0.01 (Table S3), suggesting a clear pattern of spatial aggregation. Additionally, the habitat quality of some reserves, such as Chishui Nature Reserve and Houshui River Nature Reserve in Suiyang County, transitioned from random distribution to significant spatial aggregation from 2000 to 2020, while others, including Mafulin Nature Reserve in Wuchuan County, Dashahe, and Kuankuoshui, exhibited a shift from significant spatial aggregation to random distribution.

The Getis-Ord Gi* statistical index in ArcGIS was the basis for a hotspot analysis method that was used to find habitat quality (Hotspots) and low value clustering (cold spots) in nature reserves. The interaction zones between hotspots and coldspots in nature reserves may result from developmental activities near major transportation routes and tourist facilities, leading to localized declines in habitat quality. For example, in Fanjingshan National Nature Reserve, habitat-quality hotspots areas were primarily concentrated in the core of the reserve, while coldspots areas were mainly distributed along the periphery, reflecting pressures from human activity at the periphery. In Rongjiang Moon Mountain Nature Reserve, habitat-quality hotspots showed a fragmented pattern in 2000 but evolved into a continuous distribution centered around coldspots by 2020, suggesting better habitat protection within the reserve resulting from reduced human activities (Figs. S1 to S3). Overall, from 2000 to 2020, most of the habitat-quality changes in the 33 nature reserves showed a trend of spatial aggregation that initially weakened and then strengthened. Hotspots were mainly concentrated in the core areas of the reserves, while coldspots were predominantly observed on built-up land and certain cultivated areas at the reserve boundaries.

Factors influencing the spatiotemporal evolution of habitat quality: optimal parameters-based geographical detector

Changes in habitat quality are likely driven by complex interactions of multiple factors, which, collectively, may have either a promoting or constraining effect. This study uses the bi-factor interaction detection module of the optimal parameter geographical detector model to examine how interactions between any two driving forces affect changes in habitat quality within protected areas. This method makes it possible to evaluate the interaction effects between any two elements by analyzing the interaction impacts of different influencing factors on the evolution of habitat quality. Bi-factor enhancement and non-linear enhancement are the main characteristics of interactions between any two driving factors, with no indication of separate interaction effects, according to Tables S4 and S5, which only display the top four interaction effect values.

National nature reserves

In 2000, the habitat quality of most nature reserves was predominantly influenced by natural factors, with a particularly significant impact observed in certain reserves (e.g., Fanjingshan and Guizhou Chishui Alsophila National Nature Reserve). However, the influence of natural factors in these areas gradually diminished by 2010 and 2020. This trend may indicate that the natural ecological conditions within these reserves have gradually stabilized or have benefited from effective protection measures. Meanwhile, socioeconomic factors have shown a steadily increasing influence across multiple reserves, especially in 2020 (e.g., in the Maolan and Fodingshan reserves, where the impact of land use factors has increased significantly). This trend is likely associated with socioeconomic development, population growth, and intensified human activities in areas surrounding the reserves, leading to a gradual rise in the impact of socioeconomic factors on habitat quality.

In some reserves (e.g., Leigongshan and Dashanhe), natural factors remain the dominant determinants of habitat quality, for example, elevation is a large influence among natural factors. However, in other reserves (e.g., Xishui and Kuankuoshui), the impact of socioeconomic factors has either approached or exceeded that of natural factors. This variation in influencing factors between reserves underscores the distinct environmental pressures and management challenges each reserve faces.

Although the factors influencing habitat quality vary among reserves, overall, natural factors exert a more pronounced effect on habitat quality evolution, especially geographical factors, whose explanatory power surpasses that of socioeconomic factors. Natural geographical factors provide the foundation for habitat quality evolution in national nature reserves in Guizhou Province, playing a pivotal role in either promoting or constraining the evolution of habitat quality (Fig. 5). In general, from 2000 to 2020, the influence of various factors on habitat quality within reserves has tended to stabilize or decrease.

Figure 5 Contribution of factors influencing changes in habitat quality in national nature reserves, 2000–2020.

Note: X1: elevation; X2: slope; X3: change in average annual temperature; X4: change in average annual precipitation; X5: land-use intensity; X6: gross domestic product GDP; X7: population density. y-axis indicates the magnitude of influence of the drivers.

Regional nature reserves

The habitat quality of local-level nature reserves in Guizhou Province, compared to that of national-level nature reserves, was more heavily influenced by social factors (Fig. S4). Although in some protected areas (such as R10 and R11), natural factors continue to play a dominant role, exerting a considerable influence on habitat quality, in other reserves (such as R3 and R18), socio-economic factors have a more significant impact, even surpassing natural influences. This disparity highlights the distinct characteristics of different protected areas in terms of ecological environment and external economic pressures, reflecting varied management needs among these reserves.

The interaction between natural and socio-economic factors emerged as a key driver affecting habitat quality in protected areas. The synergistic enhancement between natural and social factors became increasingly prominent over the past 20 years, with nonlinear enhancement effects being stronger than two-factor interactions. For most nature reserves, land-use intensity was the most significant factor affecting habitat quality, and its interaction with both natural and social factors was the most pronounced. Land-use intensity reflects the degree of disturbance to habitat quality caused by human development and construction activities. In protected areas, strict ecological protection often restricts economic activities by local residents, exacerbating the conflict between economic development and environmental conservation. Clearly, the influence of both land-use intensity and social factors on the natural environment is complex.

Overall, the influence of natural factors in most protected areas tends to be stable or even diminishing, while the influence of socio-economic factors is gradually increasing in many reserves. This shift may be related to regional economic development, changes in land use, and variations in management policies across protected areas.

Discussion

Frequent human activities in nature reserves can drive land-use changes, which have been identified in previous studies as a key factor influencing habitat quality (Fang et al., 2021; Wang et al., 2012). However, research on habitat quality of protected areas in Guizhou Province, including comprehensive assessments of natural geographic factors, socio-economic factors, and other anthropogenic influences, and their interaction effects, remains limited (Bai et al., 2019; Liu et al., 2023). In the present study, 33 forest ecosystem nature reserves in Guizhou Province were studied to compare habitat quality across different levels of reserves and to analyze the effects of various natural and socio-economic factors. To do this, the OPGD and InVEST models were used.

Firstly, based on the analysis of the land use transition matrix, the decline in habitat quality within protected areas is primarily attributed to environmental degradation caused by local residents aiming to increase crop yields and economic returns. This includes activities such as deforestation, grassland clearing, and shrubland conversion, which have led to varying degrees of farmland encroachment into forest, grassland, and shrubland areas, thereby contributing to habitat quality degradation. This trend aligns with observed changes in land-use data from nature reserves in Hubei Province (Lin et al., 2024). Although the overall habitat quality shows a declining trend, national nature reserves consistently exhibit higher habitat quality compared to local reserves. Previous studies, such as Jones et al. (2018)’ analysis of global nature reserves and Carranza et al. (2014)’s study of the Brazilian Cerrado, have similarly demonstrated the superior effectiveness of national reserves in habitat conservation. This can be attributed to the significant advantages of national reserves in terms of funding, resource allocation, and management standards (Zhang et al., 2017; Zhao et al., 2019). The study emphasizes the pivotal role of management levels in determining the effectiveness of habitat quality protection within protected areas. National nature reserves, with prioritized access to resources, stricter policy enforcement, and comprehensive ecological monitoring, are better equipped to mitigate anthropogenic disturbances and maintain ecological integrity. In contrast, local nature reserves, often hindered by limited management capacity and insufficient resource allocation, face greater challenges in addressing issues such as agricultural expansion and land-use conflicts, making them more vulnerable to habitat degradation. These findings highlight the necessity of tailoring conservation strategies to the specific resource needs and management capabilities of reserves at different administrative levels. The demonstrated success of national reserves underscores the potential benefits of enhancing funding, streamlining management practices, and strengthening enforcement mechanisms in local reserves. Such measures could significantly improve habitat quality protection in local reserves, contributing to more effective and comprehensive ecological conservation outcomes.

Secondly, the factors influencing habitat quality varied across different levels of nature reserves. Factor detection suggested that natural geographic factors were the primary drivers of habitat quality in national-level nature reserves (Fig. S4), given their stable and continuous influence. However, pressures from human activities remain widespread even within these nature reserves (Jones et al., 2018). For example, the Xishui and Maolan nature reserves have been influenced by the development of tourism in the surrounding regions. The growth of ecotourism and nature education programs may have driven economic activities in these areas, intensifying human-land conflicts (Zhang et al., 2023) and resulting in land-use changes that have significantly affected habitat quality. Increased tourism has been shown to exacerbate disturbance of native vegetation and wildlife (Zhang, Xiang & Li, 2012). Therefore, in these protected areas, the government needs to implement targeted measures to safeguard established objectives (Bai et al., 2019). For example, the Wolong National Natural Resource Conservation Project in Sichuan, China, is dedicated to promoting the development of eco-tourism while conserving biodiversity, achieving a win-win situation for both economic growth and environmental protection (Lu et al., 2003). This serves as an important reference for the development of protected areas in other regions.

In addition, local-level protected areas are often more significantly affected by human activity intensity, which may be partly attributed to their smaller size. For example, national-level protected areas in Northeast China tend to have stronger protective functions than local-level protected areas. Larger protected areas are generally more effective than smaller ones (Wu et al., 2022). However, in the present study, the habitat quality of Fodingshan Nature Reserve in Shibing County was observed to be primarily influenced by natural factors, likely due to its proximity to the Fodingshan National Nature Reserve. These findings emphasize the importance of coordination and interaction between national- and local-level nature reserves for an effective nature reserve system as well as for comprehensive ecosystem protection and sustainable development.

Finally, the study further reveals the joint influence of the interaction between natural and social factors on habitat quality. Over the past 20 years, the synergistic effect between the two factors has gradually strengthened and jointly driven changes in the habitat quality of nature reserves. The ranking of interaction strength indicates that, although natural factors such as topography, climate, and elevation play a fundamental supporting role in the long-term stability of habitats, social factors, including the intensity of human activities, land-use changes, and policy orientation, often have a significant impact on habitat quality in the short term and at a local scale. Notably, stringent land management policies play a critical role in maximizing the benefits of ecosystem services by mitigating the adverse effects of human activities and promoting sustainable land-use practices. This highlights the need to balance natural and social drivers in conservation planning to ensure both immediate and sustained improvements in habitat quality (Zhang, Li & Yu, 2022). Therefore, the government needs to take more measures to manage protected areas, such as regularly assessing the surrounding ecology and balancing ecological and economic development (Jabeen, Ahmad & Zhang, 2023). For example, the standardized management of ecotourism should be strengthened and policy guidance should be used to alleviate the pressure of human-land conflicts. At the same time, national-level reserves can assist local-level reserves in areas such as ecological management, resource sharing, and technical support, thereby promoting coordinated ecological protection across the region.

The findings of this study suggest that habitat quality of protected areas is influenced by interactions between natural and social factors (Lin et al., 2024). Therefore, overcoming the limitations imposed by administrative boundaries or resource classifications in geographically adjacent protected areas of the same type should be considered, and these protected areas should be merged and reorganized based on principles such as ecosystem integrity and species habitat connectivity. For example, Congjiang Moon Mountain Nature Reserve and Rongjiang Moon Mountain Nature Reserve, as well as Fodingshan National Nature Reserve and Shibing County Fodingshan Nature Reserve, could be merged allowing for a unified management. The results of the present study may also shed light on the stability of ecosystem services within these reserves, enabling a timely identification of ecosystem changes and potential challenges. Scientific support for effective management and conservation strategies is thereby provided.

In order to investigate the response mechanisms under the effect of different driving factors, including topographical features, land-use changes, and climate change, this study measures habitat quality. The results offer empirical support for a more thorough comprehension of ecological vulnerability, stability, and adaptability. The ecological pressures faced by protected areas vary across different levels of management, and habitat quality-based research supports the development of targeted management strategies to address potential impacts from activities such as tourism and agricultural expansion, thereby improving management efficiency. By systematically evaluating habitat conditions and their influencing factors, the study offers empirical evidence for formulating adaptive management policies. This approach aids in identifying and mitigating the effects of human activities on natural resources, contributing to more effective conservation and resource management practices.

It is important to acknowledge that this study is limited to habitat quality changes and influencing factors in forest ecosystem nature reserves. Future research should explore other types of nature reserves across Guizhou Province. The method used for habitat quality assessment directly affects the accuracy of the results. Assessing habitat quality over large areas remains a complex and challenging task. Currently, the assignment of correlation coefficients in the InVEST model involves a degree of subjectivity, making it difficult to validate the consistency between assessment results and actual field measurements. Moreover, field measurements of habitat quality are inherently challenging. Future research should focus on comparing the results of habitat-quality assessments based on land parcel analysis with those obtained through indicator-based field measurements, to correlate the findings from both approaches and develop a more accurate method for habitat-quality assessment.

Conclusions

This study analyzed the temporal and spatial evolution characteristics of land use and habitat quality in 33 nature reserves of forest ecosystem in Guizhou Province from 2000 to 2020, focusing on the impacts of natural and social factors on habitat quality. The results indicate that the primary land use type is forest, accounting for 90% of the total area, with land use transfers tending to be stable; however, the overall dynamism of land use has increased, suggesting relative activity. During this period, the areas of farmland, impermeable surfaces, and shrubs increased, while the areas of forest and grassland significantly decreased, and water bodies slightly declined, reflecting the negative impact of rapid urbanization on overall habitat quality. Furthermore, although habitat quality showed an overall downward trend, the rate of decline from 2010 to 2020 was less than that from 2000 to 2010, indicating some effectiveness in reserve management, with national nature reserves exhibiting better habitat quality than local ones. The spatial and temporal differentiation of habitat quality is influenced by both natural and social factors, with the interaction strength in national reserves ranked as the interaction between natural geographic factors and socioeconomic factors being greater than the internal interactions of natural geographic factors, while in local reserves, the ranking is the interaction between natural geographic factors and socioeconomic factors being greater than the internal interactions of social factors. This suggests that human activities significantly affect habitat quality and are a primary cause of habitat degradation.

Supplemental Information

Supplemental Information 1 Data

Supplemental Information 2 Analysis of cold hotspots in nationally protected areas

Supplemental Information 3 Analysis of cold spots in protected areas at the local level

Supplemental Information 4 Contribution of factors in local level protected areas

Supplemental Information 5 Land use transfer matrix for nature reserves in Guizhou Province, 2000–2020 (ha)

Supplemental Information 6 Proportion of area (%) for each habitat quality level

Supplemental Information 7 Global moran index of 33 nature reserves in Guizhou Province, 2000–2020

Supplemental Information 8 Interaction results of habitat quality factors in national nature reserves

Supplemental Information 9 Interaction results of habitat quality factors in local-level nature reserves

Additional Information and Declarations

Competing Interests

Author Contributions

Data Availability

The authors declare there are no competing interests.

Xuemeng Mei conceived and designed the experiments, performed the experiments, analyzed the data, prepared figures and/or tables, and approved the final draft.

Yi Liu performed the experiments, prepared figures and/or tables, and approved the final draft.

Li Yue conceived and designed the experiments, prepared figures and/or tables, and approved the final draft.

Mingming Zhang conceived and designed the experiments, authored or reviewed drafts of the article, and approved the final draft.

The following information was supplied regarding data availability:

The raw data is available in the Supplemental Files.

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
