# Peer review of "The spatial and temporal evolution of habitat quality and driving factors in nature reserves: a case study of 33 forest ecosystem reserves in Guizhou Province"

_PeerJ, doi:10.7717/peerj.19098_

## Round 0.1 · original submission · Major Revisions

The two reviewers find different problems, but I believe that you may be able to respond to both by clarifying the questions they have about methods and results. The paper does not have to be of global interest as stated by Reviewer 2. However I think you have an opportunity to be more didactic for international readers, by further explaining the conservation context in that region of China, what kinds of habitats are we talking about, what is the Ecological Civilisation (many people have no idea what this is about outside China), and other contextual information that can help the international reader understand the place you analysing and why improving its conservation is worth thinking about. A simple thing you can do, to start, is to put the map in the context of the map of China so we know where in China that region is located.

Reviewer 1 ·

Basic reporting

Journal PEER J
TITLE: The spatial and temporal evolution of habitat quality and its driving factors in nature reserves: a case study of 33 nature reserves of forest ecosystem in Quizhou Province (#106195)
Comments
I have the following issues
Abstract
• This is standalone section and should include briefly the basic components. This manuscript lack stating the objective; the method is incomplete; and results section must include figures/quantitative results.
2. Introduction
• This section is not well organized and discussed.
• You have discussed previously available studies on the habitat quality, but you ignore to discuss the driving force for HQ change.
• Clearly show the novelty of your study.
• The idea flows is appropriate, but indicate the specific objectives.
• You have to clearly write your specific study objectives.

3. Materials and Methods
• The manuscript is not following the journal format. For instance, 2. Materials and methods with subsection from study site description to methods of data analysis.
• Most of the equations lack citations
• How many LULC type was found in your classification?
• How did you select 33 out of the total reserves? What is the rationale behind for selecting them?
• How did you identity the drivers?
• Include the linear and exponential HQ equations.
• The source for InVEST user guide is not Romero-Calcerrada R, and Luque S. 2006.
Results
• The “Land-Use Change Analysis” needs rewriting. There are mixed ideas, which confuse readers.
• This section is not well described.
• There literature in some of them should move to the methods part.
• The results are not adequately narrated especially the HQ. Where are the results of the InVEST output?
Discussion and conclusion
The discussion section is not adequate
The conclusion part is extended and please rewrites it; focus only on your findings.
1. References are sufficient.
2. For additional and minor comments, please see the manuscript.
General comments
1. See the track change comments in the manuscript.

Experimental design

This is probably designed with issues that can be revised

Validity of the findings

No valid results that can be shared with the scientific community. The HQ using InVEST model is reliable but the results in this work is not related with model.

Additional comments

The authors vaguely explain the LULC analysis and as an experienced researcher in habitat quality, i have two critical issues :
1. The InVEST model require many inputs including the LULC maps, and the biophysical data is not clear. They list out two threats in Table 3, but in the results part they mentioned many types.
2. The HQ is expressed in values between 0 and 1, but the existed report is narration which failed to show the HQ per LULC types.

Annotated reviews are not available for download in order to protect the identity of reviewers who chose to remain anonymous.

Reviewer 2 ·

Basic reporting

In this article the authors study the spatiotemporal evolution of habitat quality in forest ecosystem of 33 nature reserves in a Chinese Province. Habitat quality is calculated using InVEST model and driving factors are obtained through geodetectors.
Overall, the article is interesting, well written and well presented. Methodologically, despite being apparently well implemented, doesn’t present any innovation. The relevance is based on the lack of such a study for the specific study area (line 90). Thus, I am afraid that it will not be of interest to an international audience since the new information is limited to the specific province. Innovation should be better argued and proven in the introduction and discussion sections. In my view, the article doesn’t do it effectively despite the potentially interesting results.
Some additional comments:
- Some important statements require reference. For example, line 95, which studies?
- Line 100, remote sensing data is mentioned but it is not clear how it was used in the study.
- The reference for the HQ model should be the original authors, and not derived studies.
- Some parts of the text need to be improved, for example, line 172 is missing a full stop and a space. There are many other situations like this throughout the document.
- Figure 1 scale should have round interval. Why use points to depict natural reserves instead of polygons? A contextual map of China should be included to help locate the study area. Why only DEM values inside the province and shaded values outside the province?
- Figure 3 is not legible. The text labels are too small.
- Figure 4, the lines of the bottom are connecting discrete variables (the natural reserves), this is not correct. What does the red dashed line mean? Needs to be described in the caption.
- Figure 5 also has legibility problems due to the small font size. Also stacked bar charts make comparison visually very difficult.
- Discussion should highlight the contribution of this to the field (and not just the province). I also find that parts of the results are not discussed being the focus mostly on factor detection.
- Conclusion could be shortened by removing the repetition of the methodology and results.

Experimental design

I think is suitable although not innovative. The used methods are quite common. Also, the study area is not likely to attract the interest of an international audience.

Validity of the findings

I think the findings are in line with the methods used and correctly implemented. This analysis does not involve accuracy assessments but that is expectect.

---

## Round 0.2 · Minor Revisions

Thank you for the thorough revisions, which have improved the clarity and communication of the paper. Reviewer 1 has some more questions or requests for clarity. I am most concerned by their points under Experimental Design, could you please try to answer these.

Perhaps I am missing something, but in your reply to reviewers you show a version of Figure 1 with an inset that shows the study area location within China. I find this useful, but I do not see it in the figure as included in the manuscript. Could you please ensure that Figure 1 includes this inset?

Reviewer 1 ·

Basic reporting

This version has good form and adequate information. Please consider the following few issues:
1. Use the origional citations for each equations
2. In the introduction part or in the method section, add concepts that clear the national and regional reserve difference in management and other issues. We are looking from the 33 reserves, 8 of them are national and the rest regional in the results section.
3. On the discussion section, we need you to associate with studies conducted some whereas.
4. In Figure 2, on the Habitat quality characteristics, there are different size maps. What are they? They are not available in the results part.

Experimental design

1. On Table 4, you included sensitivity of different LULC types threat factors. Can you include the source?
2. How did you upload the 33 reserve in the InVEST model? Using as one map or individually?
3. In the method part, state the criteria for highest, higher, ....., poor HQ using citations.

Validity of the findings

The findings are acceptable except few issues indicated in the pdf file.

Annotated reviews are not available for download in order to protect the identity of reviewers who chose to remain anonymous.

Reviewer 2 ·

Basic reporting

The authors addressed well the remarks I had raised in the previous version of the manuscript. I support its publication as is.

Experimental design

I think the experimental desing is well done.

Validity of the findings

Findings are supported on data and methods.

---

## Round 0.3 · accepted · Accept

Thank you for addressing the remaining small issues. The paper is now ready for publication.